# COUNTGD: Multi-Modal Open-World Counting

**Niki Amini-Naieni**      **Tengda Han**      **Andrew Zisserman**
Visual Geometry Group (VGG)
University of Oxford
{nikian,htd,az}@robots.ox.ac.uk

## Abstract

The goal of this paper is to improve the generality and accuracy of open-vocabulary object counting in images. To improve the generality, we repurpose an open-vocabulary detection foundation model (GroundingDINO) for the counting task, and also extend its capabilities by introducing modules to enable specifying the target object to count by visual exemplars. In turn, these new capabilities – being able to specify the target object by multi-modalites (text and exemplars) – lead to an improvement in counting accuracy. We make three contributions: *first*, we introduce the first open-world counting model, COUNTGD, where the prompt can be specified by a text description or visual exemplars or both; *second*, we show that the performance of the model significantly improves the state of the art on multiple counting benchmarks – when using text only, COUNTGD is comparable to or outperforms all previous text-only works, and when using both text and visual exemplars, we outperform all previous models; *third*, we carry out a preliminary study into different interactions between the text and visual exemplar prompts, including the cases where they reinforce each other and where one restricts the other. The code and an app to test the model are available at https://www.robots.ox.ac.uk/vgg/research/countgd/.

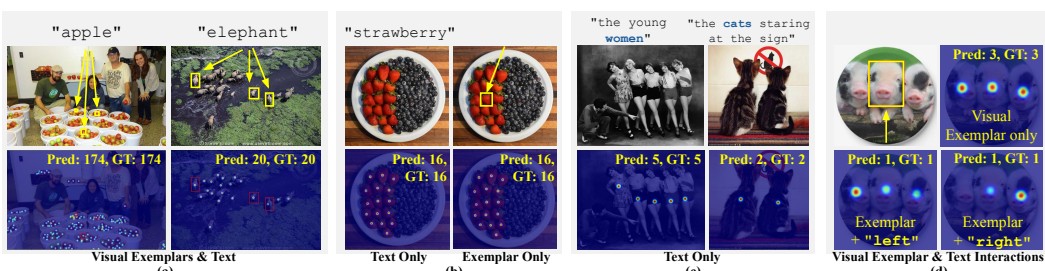

Figure 1: COUNTGD is capable of taking *both* visual exemplars and text prompts to produce highly accurate object counts **(a)**, but also seamlessly supports counting with only text queries or only visual exemplars **(b)**. The multi-modal visual exemplar and text queries bring extra flexibility to the open-world counting task, such as using a short phrase **(c)**, or adding additional constraints (the words 'left' or 'right') to select a sub-set of the objects **(d)**. These examples are taken from the FSC-147 [42] and CountBench [39] test sets. The visual exemplars are shown as yellow boxes. (d) visualizes the predicted confidence map of the model, where a high color intensity indicates a high level of confidence.

## 1 Introduction

Open-world object counting methods aim to enumerate all the instances of any category of object in an image. The 'open-world' refers to the model's ability to count objects beyond the set of categories seen at training, thus enabling the user to specify categories of interest at inference without the need for model retraining. Recent techniques allow the user to specify the target object with only visual exemplars – bounding boxes around a few example objects in the image – [32, 35], or only text

38th Conference on Neural Information Processing Systems (NeurIPS 2024).

descriptions [1, 22]. By accepting either visual exemplars or text as *prompts*, open-world object counting methods can adapt to the specific object at inference time. This enables these techniques to count arbitrary classes of objects as specified by the user.

Methods that use visual exemplars to specify the object currently significantly outperform text-based counting methods on multiple benchmarks. This is because visual exemplars provide more detailed information than text – it can take many words to precisely describe an object; and perhaps more importantly, they provide *intrinsic* information on the object's appearance – because the exemplars are from the same image they already 'factor in' the viewpoint and lighting, variables that significantly affect the object's appearance. However, while visual exemplar-based approaches are more accurate, they limit the capabilities and generality of the counting model.

In this paper we introduce a counting model that is able to specify the target object using visual exemplars, a text description, or both together. The model, named COUNTGD, has superior accuracy to previous methods, but is also more general. In addition to the performance boost obtained by specifying the target object using both visual exemplars and text, the *interaction* of the exemplars and text can be used to select a sub-set of those objects in the image. These capabilities are illustrated in Figure 1. This flexible combination of visual exemplars and text description thus provides the model with more capabilities and information than prior approaches.

To achieve this multi-modal prompt capability, we follow prior work on open-world text-specified object counting [1], and build on and extend a pre-trained vision-language foundation model, GroundingDINO [33]. We introduce new modules to embed the visual exemplars, and to enable the model to count, rather than detect. Within the model we cast the additional visual exemplars as text tokens, and the model first learns to fuse the visual exemplars with text tokens through self-attention, and then interacts with the image through cross-attention. Because the text tokens are naturally variable in length, the number of provided visual exemplars are as well. As a result, the model allows users to specify the object to count with text only, visual exemplars only, or text and any number of visual exemplars.

In summary, we make the following three contributions: *First*, we introduce COUNTGD, the first open-world object counting model that accepts either text or visual exemplars or both simultaneously, in a single-stage architecture; *Second*, we evaluate the model on multiple standard counting benchmarks, including FSC-147 [42], CARPK [20] and CountBench [39], and show that COUNTGD significantly improves on the state-of-the-art performance by specifying the target object using both exemplars and text. It also meets or improves on the state-of-the-art for text-only approaches when trained and evaluated using text-only; *Third*, we investigate how the text can be used to refine the visual information provided by the exemplar, for example by filtering on color or relative position in the image, to specify a sub-set of the objects to count. In addition we make two minor improvements to the inference stage: one that addresses the problem of double counting due to self-similarity, and the other to handle the problem of a very high count.

## 2   Related Work

Prior work on object counting has developed along three axes: (1) the density map versus detection axis, (2) the class-specific versus open-world (also referred to as "class-agnostic") axis, and (3) the visual exemplar versus text axis. The pattern is that detection, open-world, and text-based methods tend to offer more capabilities and be more general than their analogues along each axis. On the other hand, density map, class-specific, and visual exemplar-based methods tend to be more accurate at the counting tasks they apply to. COUNTGD integrates the third axis – the visual exemplar versus text axis – to achieve more general and accurate counting overall. Below, we discuss where prior work falls along each axis and where COUNTGD stands.

**Density Map versus Detection-based Object Counting** *(Axis 1).*   In the past, counting techniques that regress and sum density maps [3, 4, 7, 28, 29, 36, 46], instead of detecting and enumerating bounding boxes [6, 9, 20, 38], have proven more accurate in cluttered and dense scenes. For example, density map-based approaches like CounTX [1], LOCA [11], and CounTR [32] achieve lower counting errors than detection-based approaches such as Mask-RCNN [17] and RetinaNet [30] on standard counting benchmarks. Concurrent to our work, DAVE [40], integrates density map regression with object detection to construct a more accurate and explainable two-stage counting system. Like DAVE, COUNTGD outputs explicit object locations. However, COUNTGD is a single-stage approach that achieves better counting accuracy than DAVE and other density map-based techniques. Therefore, while density map-based approaches tend to be more accurate than detectors in highly populated

scenes, recent detection-based techniques, including COUNTGD, are beginning to achieve better accuracy than density map-based alternatives.

**Class-specific versus Open-world Object Counting** *(Axis 2).* Object counting methods first developed as class-specific techniques [4, 5, 37, 46], solving the counting problem for only one category of object, but recent methods have generalized these approaches to open-world settings, where counting *arbitrary* objects is possible. Class-specific methods have been developed to count cars [25], humans [5], and cells [14]. In contrast, open-world methods can count instances from all three categories [35]. Because class-specific techniques are more specialized than open-world approaches, they tend to be more accurate at counting instances from the class they were designed for. Recent advancements in Vision-Language Foundation Models (VLMs) such as CLIP [41] and GroundingDINO [33] trained on web-scale image-text pairs produce semantically rich visual and textual features. These features generalize to a wide range of open-world downstream tasks. Building on top of these pre-trained VLMs, recent open-world methods [1, 8, 11, 24, 32, 43, 49] have begun to surpass class-specific approaches in counting accuracy. COUNTGD, like these recent approaches, is an open-world object counter that achieves competitive performance in comparison to class-specific alternatives.

**Counting with Visual Exemplars versus Counting with Text** *(Axis 3).* Most open-world object counters approach the problem by using visual exemplars to select the objects in the input image [11, 15, 31, 32, 35, 38, 42, 43, 48, 49], but very recent work [1, 8, 22, 24, 47] has attempted to replace the visual exemplars with text, enabling new capabilities at the cost of reduced accuracy. The state-of-the-art text-based approaches, such as GroundingREC [8], CounTX [1], CLIP-Count [22], and VLCounter [24] are built on top of vision-language foundation models pretrained on large quantities of data to relate images to textual inputs and map them to a joint embedding space. This allows these foundation models to understand general concepts learned during extensive pretraining and provides a mechanism for users to specify extrinsic object properties through text. However, text-based approaches perform significantly worse than state-of-the-art visual exemplar-based approaches such as LOCA [11], CounTR [32], and few-shot DAVE [40]. For example, while both GroundingREC and COUNTGD use the pretrained GroundingDINO [33] vision-language foundation model, unlike GroundingREC, COUNTGD allows the user to input both visual exemplars and text instead of just text. This enables COUNTGD to achieve superior counting accuracy in comparison to GroundingREC. Notably, DAVE [40] is a visual exemplar-based approach that also enables textual prompts, but differs from COUNTGD in three important ways: (1) it does not address the case when both text and visual exemplars are available while COUNTGD does, (2) its comparison between text features and image features is not learned as it is by COUNTGD with attention, and (3) it is a two-stage approach, while COUNTGD solves the problem in a single stage, without relying on another visual exemplar-based counting model. Very recently, A Blind Counter (ABC) that does not require text or visual exemplars was introduced in [19]. ABC discovers different objects to count and provides exemplars indicating what has been counted. While this approach is more efficient, it does not provide the user with precise control over the object to count, as exemplar and text-based methods do.

**Relation of Counting to other areas.** Our work is related to few-shot image classification [45] and image detection [13, 21, 23] methods. These works require a few query images of novel objects, and then compare the test image with these image examples to determine its semantic content (for image classification), or to spatially localize instances (for object detection). Like these methods, COUNTGD enables us to specify the object to count with visual exemplars (i.e., "query images") but also allows for textual inputs, and then compares the test image with the multi-modal specifications to get the final count. Furthermore, we focus on the counting problem, a challenging task for object detectors.

## 3 Counting with Visual Exemplars & Text

Here, we describe COUNTGD, a single-stage model for open-world object counting that accepts either visual exemplars or text or both together as prompts to specify the object to count.

### 3.1 Overview

Given a target object specified by either visual exemplars as bounding boxes $\mathbf{B} = \{b_1, \cdots, b_N\}$ around example object instances in the image, or a textual description, $t$, or both, $\{\mathbf{B}, t\}$, the counting model, $f$, counts the number of occurrences of the object in an image $\mathbf{X} \in \mathbb{R}^{H \times W \times 3}$, as $\hat{y} = f(\mathbf{X}, \mathbf{B}, t)$, where $\hat{y}$ is the object count estimated by the counting model $f$.

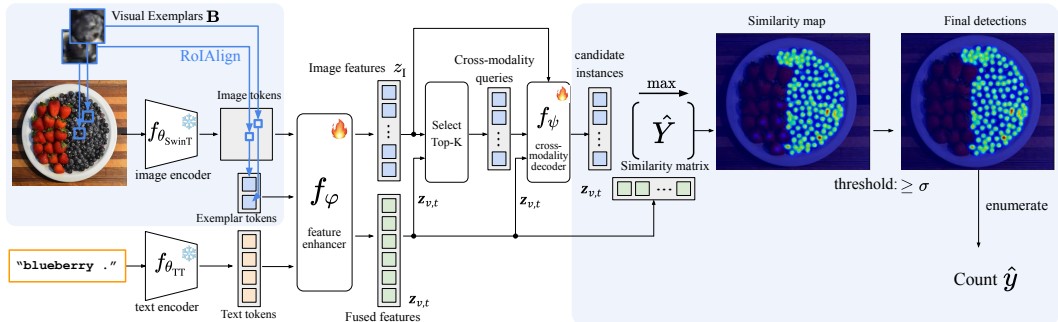

Figure 2: The COUNTGD architecture. At inference the object to be counted can be specified by visual exemplars or text prompts or both. The input image is passed through the image encoder, $f_{\boldsymbol{\theta}_{\text{SwinT}}}$ to obtain spatial feature maps at different scales. The visual exemplar tokens are cropped out of this feature map using RoIAlign (as shown in Figure 3). The text is passed through the text encoder, $f_{\boldsymbol{\theta}_{\text{TT}}}$ to obtain text tokens. In the feature enhancer, $f_{\boldsymbol{\varphi}}$, the visual exemplar tokens and text tokens are fused together with self-attention and cross-attend to the image features, producing the fused visual exemplar and text features, $\mathbf{z}_{\mathbf{v,t}}$, and new image features, $\mathbf{z_I}$. The $k$ image features $\mathbf{z_I}$ that have the highest cosine similarity with the fused features $\mathbf{z_{v,t}}$ are passed to the cross-modality decoder, $f_{\psi}$, as "cross-modality queries". Finally, the similarity matrix, $\hat{\mathbf{Y}}$ between the outputs of the cross-modality decoder, $f_{\psi}$, and $\mathbf{z_{v,t}}$ is calculated, and outputs that achieve a maximum similarity with the $\mathbf{z_{v,t}}$ above a confidence threshold $\sigma$ are identified as final detections and enumerated to estimate the final count. Our model is built on top of GroundingDINO [33] architecture with the additional modules indicated by blue shading.

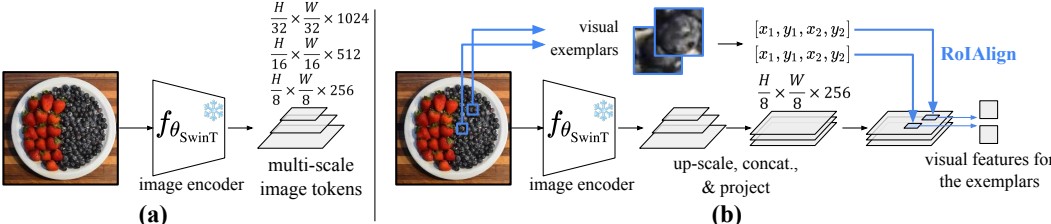

Figure 3: The visual feature extraction pipeline for images and visual exemplars. (a) For the input image, a standard Swin Transformer model is used to extract visual feature maps at multiple spatial resolutions. (b) For the visual exemplars with their corresponding bounding boxes, we first up-scale the multiple visual feature maps of the input image to the same resolution, then concatenate these feature maps, and project them to 256 channels with a $1 \times 1$ convolution. Finally, we apply a RoIAlign with the bounding box coordinates to get the visual features for the exemplars.

The architecture of the model is illustrated in Figure 2. COUNTGD is built on top of the open-world object detector GroundingDINO [33] to benefit from its pretrained open-vocabulary grounding and detection capabilities. In contrast to GroundingDINO, which only uses text queries for object detection, COUNTGD also includes visual exemplars as inputs, which increases the performance and flexibility of the model for object counting. In the following, we first describe the modules of the COUNTGD architecture, and then discuss its relation to GroundingDINO and in particular what is frozen, what is trained, and what is added to GroundingDINO.

## 3.2   COUNTGD Architecture Components

**Image Encoder** ($f_{\boldsymbol{\theta}_{\text{SwinT}}}$).    The image encoder $f_{\boldsymbol{\theta}_{\text{SwinT}}}$ encodes two types of inputs: the input image $X$ and the visual exemplars $\mathbf{B}$. The image encoder itself is the Swin-B version of the Swin Transformer [34]. As shown in Figure 3 (a), for the input image $X$, it produces spatial feature maps at three different scales. These spatial feature maps are projected to 256 dimensions with 1x1 convolutions to produce the image tokens, feature vectors of length 256 corresponding to the image patches at different scales, which are input to the feature enhancer, $f_{\boldsymbol{\varphi}}$. As shown in Figure 3 (b), for the visual exemplars $\mathbf{B}$, we reuse the spatial feature map $f_{\boldsymbol{\theta}_{\text{SwinT}}}(\mathbf{X})$ for the input image $X$, and apply aligned region-of-interest pooling, RoIAlign [16], with the pixel coordinates specified by the visual exemplars $\mathbf{B}$. The resulting visual exemplar tokens are 256-dimensional feature vectors like the image and text tokens.

**Text Encoder** ($f_{\boldsymbol{\theta}_{\text{TT}}}$).    For the text encoder, $f_{\boldsymbol{\theta}_{\text{TT}}}$, we use the BERT-base [10] text transformer pretrained on detection and phrase grounding data with the image encoder, $f_{\boldsymbol{\theta}_{\text{SwinT}}}$. The text encoder

maps the input object description $t$ to a sequence of at most 256 tokens. The encoded text tokens are 256-dimensional feature vectors. While the image encoder $f_{\theta_{\text{SwinT}}}$ produces $n$ image patch features when there are $n$ multi-scale patches extracted from the input image, and the visual exemplar encoder produces $p$ visual exemplar features when $p$ visual exemplars are available, the text encoder produces $q$ text features when there are $q$ tokens, as determined by the BERT tokenizer, in the text $t$. The $n$ image tokens, $p$ visual exemplar tokens, and $q$ text tokens are then passed to the feature enhancer $f_{\varphi}$, which fuses the three sources of information with attention.

**Feature Enhancer ($f_{\varphi}$).** The feature enhancer, $f_{\varphi}$, is composed of 6 blocks that first fuse the visual exemplar tokens with the text tokens through self-attention and then fuse the combined features with the image patch tokens with cross-attention. More specifically, each block consists of self-attention between the concatenated visual exemplar and text tokens, deformable self-attention between the image patch tokens, and image-to-text cross-attention and text-to-image cross-attention between the fused visual exemplar and text tokens and the image patch tokens. These modules enable COUNTGD to learn to relate information from the input image, visual exemplars and text query altogether. The feature enhancer $f_{\varphi}$ outputs two sets of features denoted as $\mathbf{z}_{\mathbf{v},\mathbf{t}}$ and $\mathbf{z}_{\mathbf{I}}$ as

$$(\mathbf{z}_{\mathbf{v},\mathbf{t}}, \mathbf{z}_{\mathbf{I}}) = f_{\varphi}\left(f_{\theta_{\text{SwinT}}}(\mathbf{X}), \text{RoIAlign}(f_{\theta_{\text{SwinT}}}(\mathbf{X}), \mathbf{B}), f_{\theta_{\text{TT}}}(t)\right) \qquad (1)$$

corresponding to the fused visual exemplar and text tokens, and the image patch tokens, respectively.

**Language & Visual Exemplar-guided Query Selection ($Select$).** We select the $k$ image patch tokens $\mathbf{z}_{\mathbf{I}}$ that achieve the highest similarity with the fused visual exemplar and text tokens $\mathbf{z}_{\mathbf{v},\mathbf{t}}$. This operation is denoted by $\text{Select}\left(\mathbf{z}_{\mathbf{I}}, \mathbf{z}_{\mathbf{I}}\mathbf{z}_{\mathbf{v},\mathbf{t}}^{T}, k\right)$, where $\mathbf{z}_{\mathbf{I}}\mathbf{z}_{\mathbf{v},\mathbf{t}}^{T} \in \mathbb{R}^{n \times (p+q)}$ represents the similarity scores between the $n$ image patch tokens and the $p + q$ visual exemplar and text tokens. As in GroundingDINO [33], we set $k$ to 900. These 900 image patch tokens with higher similarity scores serve as "cross-modality queries" input to the cross-modality decoder $f_{\psi}$.

**Cross-modality Decoder ($f_{\psi}$).** The cross-modality decoder, $f_{\psi}$, uses self-attention to enhance the cross-modality queries, image cross-attention to fuse the image patch features $\mathbf{z}_{\mathbf{I}}$ to the cross-modality queries, and cross-attention to fuse the visual exemplar and text features $\mathbf{z}_{\mathbf{v},\mathbf{t}}$ to the cross-modality queries. In more detail, the cross-modality decoder consists of 6 of these self-attention and cross-attention blocks. The cross-modality queries are dot-producted with the combined visual exemplar and text tokens $\mathbf{z}_{\mathbf{v},\mathbf{t}}$ and passed through an element-wise Sigmoid function to obtain the final confidence scores as:

$$\hat{\mathbf{Y}} = \text{Sigmoid}\left(f_{\psi}\left(\mathbf{z}_{\mathbf{I}}, \mathbf{z}_{\mathbf{v},\mathbf{t}}, \text{Select}(\mathbf{z}_{\mathbf{I}}, \mathbf{z}_{\mathbf{I}}\mathbf{z}_{\mathbf{v},\mathbf{t}}^{T}, k)\right)\mathbf{z}_{\mathbf{v},\mathbf{t}}^{T}\right) \qquad (2)$$

where $\mathbf{z}_{\mathbf{v},\mathbf{t}}$ are the fused visual exemplar and text features, $\mathbf{z}_{\mathbf{I}}$ are the image features, $k$ is the number of queries (i.e., maximum number of detected objects), and $\hat{\mathbf{Y}}$ are the final similarity scores that are thresholded according to a confidence threshold $\sigma$ and enumerated to estimate the final object count $\hat{y}$ at inference.

**Design choices and relation to GroundingDINO.** We choose GroundingDINO [33] over other VLMs due to its pretraining on visual grounding data, providing it with more fine-grained features in comparison to other VLMs such as CLIP [18].

To extend GroundingDINO to accept visual exemplars, we cast them as text tokens. Because both the visual exemplars and the text specify the object, we posit that the visual exemplars can be treated in the same way as the text tokens by GroundingDINO and integrate them into the training and inference procedures as such. In treating the visual exemplars as additional text tokens within a phrase, we add self-attention between the phrase corresponding to the visual exemplar and the visual exemplar rather than keeping them separate. This allows COUNTGD to learn to fuse the visual exemplar and text tokens to form a more informative specification of the object to count. Similarly, cross-attention between the image and text features in GroundingDINO's feature enhancer and cross-modality decoder becomes cross-attention between the image and the fused visual exemplar and text features in COUNTGD. Language-guided query selection in GroundingDINO becomes language and visual exemplar-guided query selection in COUNTGD. In this way, COUNTGD naturally extends GroundingDINO to input both text and visual exemplars to describe the object.

In GroundingDINO, the image encoder $f_{\theta_{\text{SwinT}}}$ is pre-trained on abundant detection and phrase grounding data with the text encoder, $f_{\theta_{\text{TT}}}$, providing it with rich region and text-aware features. Since we wish to build on this pre-trained joint vision-language embedding, we keep the image encoder $f_{\theta_{\text{SwinT}}}$ and the text encoder $f_{\theta_{\text{TT}}}$ frozen.

### 3.3 Training

We train the projection layers for extracting the visual exemplar tokens, the feature enhancer, and the cross-modality decoder of COUNTGD. The trainable parameters are updated according to a loss $\mathcal{L}$, while the rest of the parameters remain unchanged. This means COUNTGD effectively leverages the large-scale pre-training of the foundation model it extends.

The training loss $\mathcal{L}$ includes a localization term $\mathcal{L}_{loc}$ and a classification term $\mathcal{L}_{cls}$. For the localization term $\mathcal{L}_{loc}$, we regress the object centers from the final cross-modality queries output by the decoder $f_\psi$, and use the $L_1$ loss between the predicted box center $\hat{c}$ and the ground truth $c$, similar to [50]. For the classification term $\mathcal{L}_{cls}$, we compute the similarity matrix $\hat{\mathbf{Y}}$ from Equation 2 and calculate the focal loss for each score. The final loss is:

$$\mathcal{L} = \lambda_{loc}\mathcal{L}_{loc} + \lambda_{cls}\mathcal{L}_{cls} = \lambda_{loc}\sum_{i=1}^{l}|\hat{c}_i - c_i| + \lambda_{cls}\text{FocalLoss}(\hat{\mathbf{Y}}, T) \tag{3}$$

where $\lambda_{loc}$ and $\lambda_{cls}$ are hyperparameters optimized using a grid search on the validation set and $T \in \{0, 1\}^{k \times (l+1)}$ represents an optimal Hungarian matching between the $k$ predicted queries and the $l$ ground truth object instances, and the label "no object." Refer to the finetuning strategy implemented in [51] for further details.

### 3.4 Inference

To predict the object count with COUNTGD, the image $X$, text $t$, and visual exemplars $\mathbf{B}$ are inputted to the model, outputting a similarity matrix $\hat{\mathbf{Y}} \in \mathbb{R}^{k \times (p+q)}$. The maximum score over all $p + q$ visual exemplar and text tokens is extracted for each of the $k$ queries. Maximum scores above a confidence threshold $\sigma$ are enumerated to estimate the object count.

## 4 Experiments

COUNTGD is trained on the FSC-147 [42] object counting dataset training set, and then evaluated on the FSC-147 test set, and two other benchmark datasets (without any fine-tuning). We first describe the datasets, and then discuss the performance.

### 4.1 Datasets & Metrics

**FSC-147 [42].** FSC-147 contains 6135 images with 89 classes in the training set, 29 classes in the validation set, and 29 classes in the test set. The classes in the training, validation, and test sets do not overlap. Each image is annotated with at least three visual exemplars. For text descriptions, we use the singular forms of the class names in FSC-147-D [1] with any prefixes such as "the" removed. For example, we change "the donuts in the donut tray" in FSC-147-D to "donut" by removing the prefix "the," extracting the class name "donuts," and then singularizing it to "donut."

*Corrections to FSC-147*. We make two corrections to FSC-147 and report results with and without these corrections. (1) As noted in [32], image 7171.jpg has incorrect visual exemplars labeled. Since, unlike the model in [32], COUNTGD can input either visual exemplars or text, for this example we only provide the model with text. (2) Image 7611.jpg has the incorrect text description "lego" even though the lego *studs* not the lego *bricks* should be counted. We change the description to "yellow lego stud" for this example.

**CARPK [20].** CARPK contains images of parking lots captured by overhead drones with a training set and test set of 989 and 459 images respectively. Each image is annotated with at least two bounding boxes. We use the same two bounding boxes selected in [32] as the visual exemplars for each image. We use the class name "car" as the text description.

**CountBench [39].** CountBench contains 540 images with 2-10 objects and captions describing the image as well as the number of objects to count. We create text descriptions for a 504-image subset of CountBench, removing inappropriate images and images with links that are unavailable. We give details of how the class names are obtained from the captions accompanying each image in the Appendix.

**Metrics.** Following prior work on object counting [1, 11, 32], the Mean Absolute Error (MAE) and the Root Mean Squared Error (RMSE) are used to measure performance. We define these metrics in the Appendix.

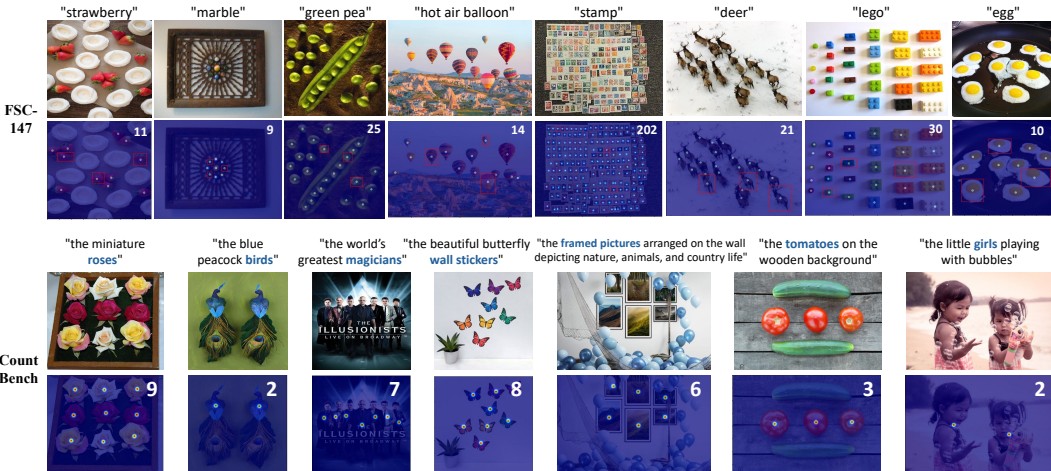

Figure 4: Qualitative counting results on FSC-147 [42] and CountBench [39] using the multi-modal COUNTGD. The model is trained and tested on FSC-147 visual exemplars and text. Input text is written above each image, and visual exemplars are indicated by the red boxes. On CountBench, we test the same model trained on the FSC-147 in a zero-shot way with only text (there are no visual exemplars for CountBench). Blue words indicate the subject of each caption input to the model. In both cases, COUNTGD predicts the count in all images shown with 100% accuracy. Note on the CountBench examples, the model counts the specified objects correctly when there are multiple types of objects in the image, such as the tomatoes with cucumbers, and the girls with bubbles. Detected points are filtered with a Gaussian and plotted under the input images for visualization purposes.

## 4.2 Implementation

**Training.** The model is trained for 30 epochs on the FSC-147 training dataset using Adam optimizer and standard augmentations. The image and text encoders, $f_{\theta_{\text{SwinT}}}$ and $f_{\theta_{\text{TT}}}$, are frozen during training. Full details are given in the Appendix.

**Inference.** At inference, each image is resized such that its shortest side length is 800 pixels, and its aspect ratio is maintained. The image is then normalized and passed to the model. The visual exemplars are passed in as bounding boxes, and the special token " ." is appended to the text description before providing it to the model. In the Appendix we give details of two important improvements: one to avoid double counting given self-similarity of the target object (like a butterfly [32]), and the other using adaptive cropping to overcome the 900 counting quota of the the model.

## 4.3 Comparison to State-of-the-art on Standard Benchmarks

Here we show that COUNTGD achieves comparable or exceeds state-of-the-art performance for text-only open-world object counting when using only text, and exceeds the performance of all open-world object counting methods when using both visual exemplars and text on three benchmarks.

**FSC-147 [42].** In Table 1, we test COUNTGD under two settings: (1) trained and tested with only text (denoted as COUNTGD$_{\text{txt}}$), and (2) trained and tested with both 3 visual exemplars and text (denoted as COUNTGD). COUNTGD trained and tested with both visual exemplars and text sets a new state-of-the-art for counting accuracy on FSC-147, achieving significantly lower counting errors than all prior approaches to open-world object counting. Training with only text achieves comparable counting accuracy to state-of-the-art text-only open-world object counting methods. The concurrent method GroundingREC [8] achieves slightly lower mean absolute error values than COUNTGD$_{\text{txt}}$, while COUNTGD$_{\text{txt}}$ achieves lower root mean squared error values. The results for GroundingREC and COUNTGD$_{\text{txt}}$ are likely close to each other since both methods leverage the pretrained GroundingDINO [33] foundation model. However, unlike GroundingREC, COUNTGD can fuse information from both text and visual exemplars instead of using only text, enabling a significant improvement. Similarly, while a pre-trained GroundingDINO performs poorly at counting (top row of Table 1), a GroundingDINO model fine-tuned on FSC-147 achieves good results [8], that match the performance of COUNTGD$_{\text{txt}}$. Adding visual exemplars to COUNTGD significantly improves its performance over fine-tuned GroundingDINO (Table 1, lowest row shows a test MAE of

Table 1: FSC-147 [42] comparison with the state-of-the-art text-only and visual exemplar-only open-world counting methods. Multi-modal COUNTGD trained and tested with both visual exemplars and text achieves state-of-the-art counting accuracy for open-world object counting, beating all text-only and visual exemplar-only approaches. COUNTGD_{txt} trained and tested with only text achieves comparable performance to state-of-the-art text-only counting approaches. * = correction of erroneous GT labels, as explained in section 4.1. GroundingREC [8], DAVE_{prm}, and DAVE [40] are concurrent work. *Lower MAE and RMSE values mean more accurate results.*

| Method | Year | Paper Venue | How to Specify the Class | Validation MAE ↓ | Validation RMSE ↓ | Test MAE ↓ | Test RMSE ↓ |
|---|---|---|---|---|---|---|---|
| GroundingDINO [33] | 2024 | ECCV | Text | 54.45 | 137.12 | 54.16 | 157.87 |
| Patch-selection [47] | 2023 | CVPR | Text | 26.93 | 88.63 | 22.09 | 115.17 |
| CLIP-count [22] | 2023 | ACMMM | Text | 18.79 | 61.18 | 17.78 | 106.62 |
| VLCounter [24] | 2023 | AAAI | Text | 18.06 | 65.13 | 17.05 | 106.16 |
| CounTX [1] | 2023 | BMVC | Text | 17.10 | 65.61 | 15.88 | 106.29 |
| CounTX* [1] | 2023 | BMVC | Text | 17.10 | 65.61 | 15.69 | 106.06 |
| DAVE_{prm} [40] | 2024 | CVPR | Text | 15.48 | 52.57 | 14.90 | 103.42 |
| GroundingREC [8] | 2024 | CVPR | Text | **10.06** | 58.62 | **10.12** | 107.19 |
| COUNTGD_{txt} (ours) | 2024 | NeurIPS | Text | 12.14 | 47.51 | 14.76 | 120.42 |
| COUNTGD$^*_{txt}$ (ours) | 2024 | NeurIPS | Text | 12.14 | **47.51** | 12.98 | **98.35** |
| CounTR [32] | 2022 | BMVC | Visual Exemplars | 13.13 | 49.83 | 11.95 | 91.23 |
| LOCA [11] | 2023 | ICCV | Visual Exemplars | 10.24 | 32.56 | 10.79 | 56.97 |
| DAVE [40] | 2024 | CVPR | Visual Exemplars | 8.91 | 28.08 | 8.66 | 32.36 |
| COUNTGD (ours) | 2024 | NeurIPS | Visual Exemplars & Text | 7.10 | 26.08 | 6.75 | 43.65 |
| **COUNTGD* (ours)** | **2024** | **NeurIPS** | **Visual Exemplars & Text** | **7.10** | **26.08** | **5.74** | **24.09** |

Table 2: Comparison with state-of-the-art open-world counting methods. **(top)** On CARPK [20], we compare with text-only and visual exemplar-only methods. COUNTGD, trained with both visual exemplars and text on FSC-147 [42], achieves lower error values than all text-only and visual exemplar-only methods, without being trained on any images in CARPK, using either text-only or both text and two visual exemplars at inference. **(bottom)** On CountBench [39], we compare with currently the best publicly available text-only open-world counting method, CounTX [1]. COUNTGD (trained on both visual exemplars and text), given only text and zero-shot, achieves significantly lower errors than CounTX. Note, CountBench does not provide visual exemplars.

| Dataset | Method | Year | Paper Venue | How to Specify the Class | Fine-tuned | Test MAE ↓ | Test RMSE ↓ |
|---|---|---|---|---|---|---|---|
| CARPK | CLIP-count [22] | 2023 | ACMM | Text | ✗ | 11.96 | 16.61 |
| | CounTX [1] | 2023 | BMVC | Text | ✓ | 8.13 | 10.87 |
| | VLCounter [24] | 2023 | AAAI | Text | ✗ | 6.46 | 8.68 |
| | **COUNTGD (ours)** | **2024** | **NeurIPS** | **Text** | ✗ | **3.83** | **5.41** |
| | LOCA [11] | 2023 | ICCV | Visual Exemplars | ✗ | 9.97 | 12.51 |
| | CounTR [32] | 2022 | BMVC | Visual Exemplars | ✓ | 5.75 | 7.45 |
| | SAFECount [49] | 2022 | WACV | Visual Exemplars | ✓ | 5.33 | 7.04 |
| | **COUNTGD (ours)** | **2024** | **NeurIPS** | **Visual Exemplars & Text** | ✗ | **3.68** | **5.17** |
| CountBench | CounTX [1] | 2023 | BMVC | Text | ✗ | 6.64 | 15.75 |
| | **COUNTGD (ours)** | **2024** | **NeurIPS** | **Text** | ✗ | **0.86** | **3.1** |

5.74 and a test RMSE of 24.09 for COUNTGDcompared to the test MAE of 10.82 and test RMSE of 104 noted in [8] for fine-tuned GroundingDINO). Unlike COUNTGD, GroundingDINO does not allow for visual exemplars as additional inputs.

In Figure 4, we give qualitative examples of the detections that COUNTGD outputs given both visual exemplars and text from the FSC-147 test set. Note how in the first image, COUNTGD only counts the strawberries and not the white cookies. Prior work has shown that visual exemplar-only methods struggle to count only one category of object when there are repeating instances from multiple categories in an image [40]. COUNTGD handles this issue very well in this example by leveraging the generalization capabilities of the pretrained vision-language model GroundingDINO [33].

**CARPK [20].** To test cross-dataset generalization, COUNTGD is trained on FSC-147 [42], and tested on the CARPK car counting dataset zero-shot, without being trained on any images in CARPK. In Table 2, COUNTGD is trained on the FSC-147 [42] training set with both visual exemplars and text, and tested on the CARPK car counting dataset under two settings: (1) using only the text input "car", and (2) using both the text input "car" and the same two visual exemplars as [32]. Under both settings, COUNTGD achieves state-of-the-art accuracy on CARPK for all open-world object counting methods without being trained on any images in CARPK, achieving lower counting errors than methods like CounTR [32] and SAFECount [49] that were fine-tuned on CARPK.

**CountBench [39].** We train COUNTGD on FSC-147, which has at least seven objects in each training image, and evaluate its generalization to counting low numbers of objects in the CountBench test set zero-shot. In Table 2, we compare COUNTGD's performance on counting low numbers

Table 3: Ablation study I: CₒᴜɴᴛGD trained and tested with text only, visual exemplars only, and text and visual exemplars together on FSC-147 [42]. Multi-modal CₒᴜɴᴛGD trained and tested with both text and visual exemplars achieves the lowest counting errors.

| Training and testing setting | Val | | Test | |
|---|---|---|---|---|
| | MAE ↓ | RMSE ↓ | MAE ↓ | RMSE ↓ |
| CₒᴜɴᴛGD (Text) | 12.14 | 47.51 | 12.98 | 98.35 |
| CₒᴜɴᴛGD (Visual Exemplars) | 7.46 | 29.54 | 8.31 | 91.05 |
| **CₒᴜɴᴛGD (Text & Visual Exemplars)** | **7.10** | **26.08** | **5.74** | **24.09** |

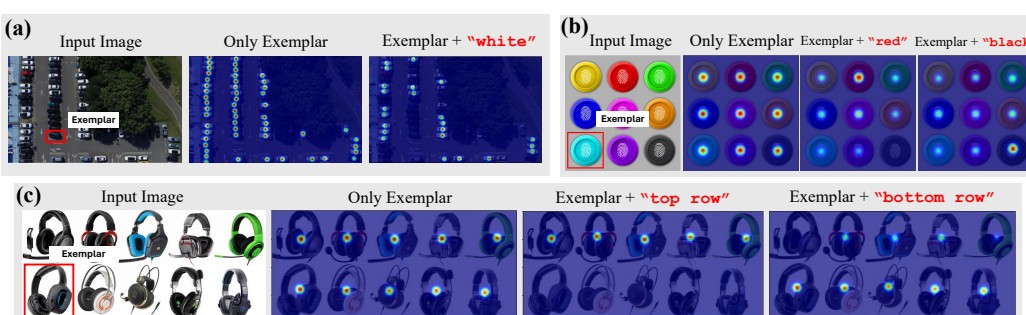

Figure 5: Studying visual exemplar and text interactions. We plot the confidence scores of the instances for each image. In (a) and (b) we show we can specify shape with the exemplar and modify color with text. In (c) we show we can specify spatial location with text, and shape with the exemplar.

of objects (2–10) to CounTX [1], currently the best (according to performance on FSC-147 [42]) publicly available pre-trained open-world text-specified object counting methods. For this experiment, CₒᴜɴᴛGD trained with both visual exemplars and text on FSC-147, is tested on CountBench zero shot given only text. Because CountBench contains long captions that describe more than the object to count, we only threshold text token similarity scores corresponding to the subject of each caption. CₒᴜɴᴛGD achieves significantly better performance than CounTX on this dataset. In Figure 4, we show qualitative examples of the detections output by CₒᴜɴᴛGD. The subject of each caption is shown with yellow text.

## 4.4 Ablation Study

**Uni-Modal vs. Multi-Modal Training.** In Table 3, we compare CₒᴜɴᴛGD's performance using different training and inference procedures on FSC-147 [42]. Training on text only and testing with text only achieves performance comparable to state-of-the-art counting accuracy for text-only approaches, demonstrating the superiority of the GroundingDINO [33] architecture that we leverage. Training with visual exemplars only and testing with visual exemplars only results in state-of-the-art performance on two out of four of the metrics (mean absolute errors on both the validation and test sets) for visual exemplar-only approaches. This is surprising given that GroundingDINO was pretrained to relate text to images not visual exemplars to images. Despite this, CₒᴜɴᴛGD performs remarkably well in this setting. Multi-modal training and testing with both visual exemplars and text beats both uni-modal approaches and sets a new state-of-the-art for open-world object counting. This ablation study shows that the visual exemplars provide more information than the text in FSC-147 as the performance with visual exemplars only is significantly better than the performance with text only. It also demonstrates that multi-modal training and inference is the superior strategy as it allows CₒᴜɴᴛGD to take advantage of two sources of information about the object instead of one. In Table 5 in the Appendix, we additionally include an ablation study showing the influence of our proposed SAM Test-time normalization and adaptive cropping strategies.

## 4.5 Language and Exemplar Interactions

Up to this point we have used the text and visual exemplar prompts to specify the target object in a complementary manner; for example giving a visual exemplar of a 'strawberry' with the text 'strawberry'. It has been seen that the counting performance with prompts in both modalities is, in general, equal or superior to text alone. In this section we investigate qualitatively the case where the text refines or filters the visual information provided by the exemplars. For example, where the visual exemplar is car, but the text specifies the color, and only cars of that color are counted.

In this study, unlike before, we freeze the feature enhancer in addition to the image and text encoders and finetune the rest of the model on FSC-147 [42]. We find that freezing the feature enhancer is necessary for many of these interactions to emerge. Once trained, the new model can use the text to filter instances picked out by the exemplar, and the exemplar can increase the confidence when it reinforces the text. In Figure 5 we show several examples of the interactions observed.

## 5 Conclusion & Future Work

We have extended the generality of open-world counting by introducing a model that can accept visual exemplars or text descriptions or both as prompts to specify the target object to count. The complementarity of these prompts in turn leads to improved counting performance. There are three research directions that naturally follow on from this work: (i) the performance could probably be further improved by training on larger scale datasets, for example using synthetic data as demonstrated recently for counting [27]; (ii) a larger training set would enable a thorough investigation of freezing more of the GroundingDINO model when adding our new visual exemplar modules; and finally, (iii) the model does not currently predict the errors of its estimates like other computer vision models do [2, 44].

**Acknowledgement**

The authors would like to thank Shilong Liu for his extensive support of GroundingDINO [33], Zechen Bai for his extensive support of Patch CLIP introduced in [12], and Kiana Amini-Naieni for her help in labeling the CountBench [39] images. We would also like to thank Oishi Deb, Abhishek Dutta, Horace Lee, Orest Kupyn, Vladimir Iashin, and Paul Engstler for providing detailed feedback on the CountGD app. We would further like to thank Ahsen Khaliq, Toshihiro Hayashi, Jean-Benoit Delbrouck, Luc Georges, Michelle Habonneau, and Yuvraj Sharma for their help in deploying the app on Hugging Face. The app is supported by a Hugging Face Community GPU grant. This research is funded by an AWS Studentship, the Reuben Foundation, the AIMS CDT program at the University of Oxford, EPSRC Programme Grant VisualAI EP/T028572/1, and a Royal Society Research Professorship RP\R1\191132.

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

# Appendix

## A    Definition of Metrics

We use the Mean Absolute Error (MAE) and the Root Mean Squared Error (RMSE) to measure performance. They are defined as:

$$\text{MAE} = \frac{1}{N} \sum_{i=1}^{N} |\hat{y}_i - y_i|, \quad \text{RMSE} = \sqrt{\frac{1}{N} \sum_{i=1}^{N} (\hat{y}_i - y_i)^2} \tag{4}$$

where $N$ is the number of test images, $\hat{y}_i$ is the predicted count, and $y_i$ is the ground truth count for image $X_i$. For both MAE and RMSE, a lower value indicates a better performance.

## B    Additional Dataset Details

**CountBench [39]**    Here we explain how the descriptions and keywords for CountBench were constructed. Unlike the original CountBench captions, our text descriptions include the object to count without revealing the number of objects. For example, the caption "background photo of three light bulbs" in CountBench is replaced with "the light bulbs," which describes the object to count (the light bulbs) without giving away that there are three in the image. Because some descriptions include information about other objects in the image, we add keywords that indicate the subject in the caption. For example, the caption "the children standing on a bench at an outdoors party" includes the keyword "children" to indicate that the children, not the bench, should be counted. Providing keywords is necessary since COUNTGD has been pretrained on visual grounding data and will count both the children and the bench they are sitting on as a result. To ensure only the children are counted, text token similarity scores from the keyword "children" are thresholded to estimate the count. Because CountBench contains very few objects, we do not use visual exemplars and provide COUNTGD with just the text description.

## C    Additional Implementation Details

**Architecture.**    We provide additional architectural details here. The image encoder, $f_{\boldsymbol{\theta}_{\text{SwinT}}}$, is a Swin-B transformer with corresponding patch sizes $8 \times 8$, $16 \times 16$, and $32 \times 32$ and final embedding dimensions of 192, 384, and 768 respectively. To get the visual exemplar tokens, the spatial feature maps from $f_{\boldsymbol{\theta}_{\text{SwinT}}}(\mathbf{X})$ are first upsampled to the same height and width as the largest one with patch size $8 \times 8$. The upsampled feature maps are concatenated along the channel dimension and projected to 256 dimensions with a separate $1 \times 1$ convolution. RoIAlign is then applied to extract features corresponding to the exemplar regions,

Table 4: Sensitivity of CountGD's counting accuracy to $\boldsymbol{\lambda_{loc}} : \boldsymbol{\lambda_{cls}}$ on FSC-147 given *both* text and exemplars. Decreasing $\boldsymbol{\lambda_{loc}}/\boldsymbol{\lambda_{cls}}$ improves the val. errors more than increasing it does. Deviating $\boldsymbol{\lambda_{loc}}/\boldsymbol{\lambda_{cls}}$ from one worsens the test errors with increasing it harming the test accuracy the most. We choose $\lambda_{loc} : \lambda_{cls} = 1 : 5$ as this achieves the lowest validation set MAE.

| $\lambda_{loc}$ | $\lambda_{cls}$ | Val. | | Test | |
|---|---|---|---|---|---|
| | | MAE ↓ | RMSE ↓ | MAE ↓ | RMSE ↓ |
| 1 | 1 | 8.64 | 44.71 | **5.62** | **21.58** |
| 5 | 1 | 8.55 | 35.65 | 8.01 | 82.55 |
| **1** | **5** | **7.10** | **26.08** | 5.74 | 24.09 |

**Training.**    Each training image is first horizontally flipped with a probability of $50\%$. Next, with a probability of $50\%$, either the minimum side length of the image is resized to a side length in $\{480, 512, 544, 576, 608, 640, 672, 704, 736, 768, 800\}$ such that the aspect ratio of the image is maintained or the image is first randomly cropped such that the minimum side length is in the range $[384, 600]$ and then resized as mentioned before. After this, the image is normalized and passed through the model. Following [33], all the classes in the FSC-147 training set are concatenated into a single caption with " ." separating each class name. Visual exemplar tokens are appended to the end of the text tokens associated with their class name. Self-attention masks are constructed such that the

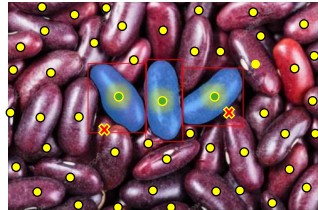 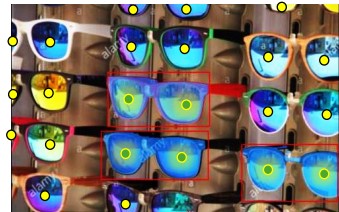

(a) CounTR [32] incorrectly detects self-similarity.   (b) SAM TT-Norm correctly detects self-similarity.

Figure 6: In example (a), the TT-Norm presented in [32] would incorrectly detect self-similarity since multiple correctly detected instances, denoted by red '×'s, fall within the exemplar regions, indicated by the red boxes. However, the blue segmentation masks output by SAM only contain one detected instance per mask, so the SAM TT-Norm correctly does not detect self-similarity. (b) shows a case where the SAM TT-Norm correctly detects self-similarity and divides the estimated count by 2.

text tokens attend to each other as well as to the visual exemplars that are associated with their class name. Self-attention is not applied between unrelated class names and visual exemplars. The model is optimized with the Adam Optimizer with a weight decay set to $10^{-4}$ and an initial learning rate set to $1 \times 10^{-4}$ that reduces by a factor of ten every ten epochs. $\lambda_{loc}$ is set to 1 and $\lambda_{cls}$ is set to 5 in Equation 3. These scale factors are also used in the Hungarian Matching Cost for matching ground truth points to predicted points. The confidence threshold $\sigma$ is set to 0.23. Hyperparameters are set using default values provided by [51] with the exception of selecting $\lambda_{loc}$, $\lambda_{cls}$, and the confidence threshold $\sigma$ using a sparse grid search optimizing the mean absolute counting error on the validation set. Specifically $(\lambda_{loc}, \lambda_{cls}) \in \{1, 2.5, 5\} \times \{1, 2.5, 5\}$ and $\sigma \in \{0.14, 0.17, 0.2, 0.23\}$ are tested. In Table 4, we conduct a sensitivity test showing that our choice of $(\lambda_{loc}, \lambda_{cls}) = (1, 5)$ achieves the best results for the validation set. Given these parameters, our choice of $\sigma = 0.23$ achieves the best validation set MAE of 7.1, while $\sigma = 0.14$ achieves the worst validation set MAE of 9.5. The model is trained for 30 epochs with early stopping with respect to the mean absolute counting error on the validation set with no SAM TT-Norm or adaptive cropping applied. We train the multi-modal model that beats all prior methods for open-world object counting using both visual exemplars and text. We train the text-only model that beats all text-only approaches to open-world object counting on only text. Finally, we additionally freeze the feature enhancer for the exemplar and text interaction study.

**Training Resources.**   Our model is trained on 1 Nvidia A6000 GPU with 48GB of graphic memory. A full training takes about 1 day.

## D   Additional Inference Details

**Avoiding double counting.**   One of the common problems for counting models is handling self-similarity, when an object is intrinsically repetitive. For example, sunglasses and butterflies exhibit self-similarity. In these cases, counting methods tend to double count, detecting each self-similar component. CounTR [32] has tried to address this by dividing the estimated count by the average count in the visual exemplar regions. We observe that this approach (referred to as the "TT-Norm") fails in cluttered scenes, where visual exemplar bounding boxes encapsulate more than just one instance of the object. This causes the counting model to detect self-similarities when they are not present. We show an example of this in Figure 6 (a).

To address this, we propose to use segmentation masks instead of bounding boxes to more accurately check if the counting model detects more than one instance on an object inside a visual exemplar. To obtain the segmentation masks we use the visual exemplars as box prompts to the Segment Anything Model (SAM) [26]. This approach avoids the issue faced by CounTR's TT-Norm since we do not check instances outside the object's boundary, even if these instances fall within the visual exemplar regions. Figure 6 shows how our approach compares to CounTR's TT-Norm in such cases.

**Adaptive Cropping.**   We address the problem that COUNTGD can only output at most 900 queries at a time through adaptive cropping. If COUNTGD detects 900 objects, we crop the input image into

Table 5: Ablation study II: CountGD is tested with different inference procedures on FSC-147 [42]. TT-Norm refers to test-time normalization. Correction (1) refers to using only text for image 7171.jpg, and Correction (2) refers to correcting the incorrect text description for image 7611.jpg from "lego" to "yellow lego stud."

| SAM TT-Norm | Adaptive Cropping | Correction (1) | Correction (2) | Val | | Test | |
|---|---|---|---|---|---|---|---|
| | | | | MAE ↓ | RMSE ↓ | MAE ↓ | RMSE ↓ |
| ✗ | ✗ | ✗ | ✗ | 8.69 | 43.89 | 10.92 | 99.58 |
| ✓ | ✗ | ✗ | ✗ | 7.99 | 42.23 | 9.62 | 98.90 |
| ✓ | ✓ | ✗ | ✗ | 7.10 | 26.08 | 6.75 | 43.65 |
| ✓ | ✓ | ✓ | ✗ | 7.10 | 26.08 | **5.70** | **24.04** |
| ✓ | ✓ | ✓ | ✓ | **7.10** | **26.08** | 5.74 | 24.09 |

Table 6: Ablation Study III: The SAM TT-Norm provides a small improvement to the counting accuracy. Adaptive cropping is applied here. Note: CountGD still achieves state-of-the-art accuracy without the SAM TT-Norm.

| SAM TT-Norm | Val | | Test | |
|---|---|---|---|---|
| | MAE ↓ | RMSE ↓ | MAE ↓ | RMSE ↓ |
| ✗ | 7.79 | 28.70 | 7.03 | 26.74 |
| ✓ | **7.10** | **26.08** | **5.74** | **24.09** |

multiple overlapping pieces and pass each cropped piece back to the model. The crop width and height are calculated as 4 times the average exemplar width and height. This approximately upper bounds the number of objects that can appear inside the crop window. The overlap width and height are determined to be 1.25 times the average exemplar width and height to approximately ensure each object instance appears fully in at least one crop. If visual exemplars are not provided, the image is cropped into four equal pieces. To obtain the final count, the number of detected instances in each crop window are added together while averaging the predicted count in overlapping regions.

**Ablation Study.** In Table 5, we test the influence of the SAM TT-Norm, Adaptive Cropping, and our two corrections to the FSC-147 annotations on the multi-modal CountGD model. The SAM TT-Norm provides minor improvements. This is because only a small number of classes in FSC-147 exhibit self-similarity. The adaptive cropping provides significant improvements with respect to the RMSE and minor improvements with respect to the MAE. This is because the adaptive cropping is specifically for handling high counts of objects ($\geq 900$), and the RMSE is particularly sensitive to errors from these outliers. Correction (1), only using text for image 7171.jpg, has a significant influence on the RMSE as using the incorrectly annotated visual exemplars for this example causes CountGD to correctly count the lego studs identified by the exemplars, not the lego bricks, and there are a high number of studs and a small number of bricks. Providing only the text "lego" and discarding the erroneous visual exemplars fixes this issue. Correction (2), correcting the text description for 7611.jpg, has no significant influence on the multi-modal model.

In Table 6, we show that by adding adaptive cropping back in (unlike in Table 5, rows 1-2), CountGD still achieves state-of-the-art counting accuracy even without the test-time normalization.

To further investigate the effectiveness of the adaptive cropping, we split images into different groups according to the number of objects they contain, and report the percent error for each group. We find that for FSC-147 test images with more than 900 objects, the mean percent error ($\frac{|gt-pred|}{gt} \times 100\%$) is 10%, and for images with at most 900 objects, it is 8%. This shows that CountGD works well, even for images with greater than 900 objects.

# E   One-Shot Counting

Table 7: CountGD's performance on 1-shot versus few-shot counting on FSC-147. Providing more exemplars in the prompt increases the accuracy of CountGD.

| Inference Setting | Val | | Test | |
|---|---|---|---|---|
| | MAE ↓ | RMSE ↓ | MAE ↓ | RMSE ↓ |
| 1 exemplar + text | 8.00 | 30.29 | 8.7 | 83.21 |
| **3 exemplars + text** | **7.10** | **26.08** | **5.74** | **24.09** |

In Table 7, we consider how well COUNTGD performs given text and only one visual exemplar, instead of text and three visual exemplars as in the last row of Table 1. As expected, increasing the number of visual exemplars improves the accuracy of the estimated count.

## F  Training With Different Ratios of Text and Visual Exemplars

Table 8:  CountGD's performance on FSC-147 using different ratios of text versus exemplar training data. Providing more exemplar training data results in better performance overall, when given text only, exemplars only, or both at inference.

| Training Data Distribution | Test MAE ↓ | | |
| --- | --- | --- | --- |
| | Text only | 3 Exemplars Only | Text + 3 Exemplars |
| 80 % Text-only, 20 % Exemplar-only | 16.98 | 7.66 | 8.14 |
| 20 % Text-only, 80 % Exemplar-only | **12.67** | **6.28** | **5.51** |

Although we train multi-modal COUNTGD *jointly* on both text and exemplars (it is given one text description and 3 exemplars for each training image simultaneously), for this experiment we consider how training with different ratios of exemplar-only and text-only data affects the counting performance. Specifically, we train COUNTGD in two settings. For the first setting (first row of Table 8), we drop the exemplars $80\%$ of the time and train with only text. For the remaining $20\%$ of the time, we train with only exemplars and drop the text. For the second setting (second row of Table 8), we drop the text $80\%$ of the time and train with only exemplars, training with only text for the remaining $20\%$ of the time.

As shown in Table 8, training with more exemplar-only data improves performance, even when testing with text only. We hypothesize this may be because training on more exemplar-only data regularizes COUNTGD's text-only performance. COUNTGD is initialized with the pretrained GroundingDINO [33] weights, so it already starts with a strong understanding of how text and images relate to each other. Perhaps limiting the amount of text-only training data preserves this understanding obtained through large-scale pretraining. On the other hand, GroundingDINO has not been pretrained to relate the exemplars to the image, and, thus, benefits from having more exemplar-only training data to learn this capability.

## G  Diving Deeper Into Language and Exemplar Interactions

In Section 4.5 we discuss a preliminary study on the interactions that emerge between the exemplars and the text. In this study, we plot the average confidence score over both the exemplar and the text tokens. When the exemplar and the text reinforce each other, the average confidence tends to increase overall.

Differently, here, we carry out a preliminary investigation into what happens when the exemplar and the text *conflict* with each other. We define the desired behavior to be an AND operation between the exemplar and text tokens, where correctly counted objects should match *both* the exemplar and the text. Since this is not possible when the two conflict with each other, we expect an ideal count to be zero. We investigate to what extent this holds in practice by defining three levels of conflict.

1. super-class conflict – the super class of the exemplar and the text don't match e.g., the exemplar is a tiger and the text='chair';

2. sub-class conflict – the sub-class of the exemplar and text don't match e.g., the exemplar is a man and the text='woman', both of which are humans;

3. attribute conflict – the exemplar and text match in terms of class but don't match in terms of attribute, e.g., the exemplar is a blue circle but the text='red'

For case (1) we use an image of the butterflies from CountBench [39]. By providing visual exemplars of the butterflies and the text 'white plant pot,' we get a count of 0 as expected. For case (2) we use an image of the strawberries and blueberries from FSC-147 [42]. By providing one exemplar of a blueberry and the text 'strawberry', we obtain a count of 0. For case (3), we consider colored roses in an image from CountBench. In this case, when providing an exemplar of a red rose and the text 'yellow,' the output is (incorrectly) 9, the number of red and yellow roses. We speculate that we are

limited by image-text capabilities of the original GroundingDINO [33] model (as illustrated in the fine-grained limitation example provided in Appendix I). We include qualitative results from this experiment in Figure 7.

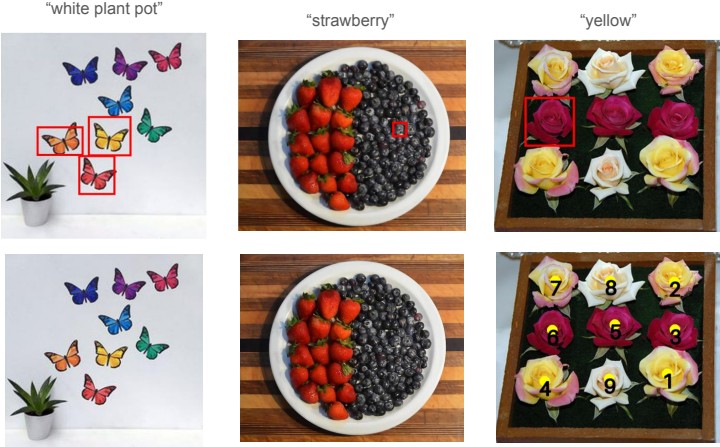

Figure 7: Visualizing CountGD's output when the exemplar and text conflict. In the top row we show the input image, text input, and visual exemplars provided to COUNTGD. In the bottom row, we visualize COUNTGD's output. For the butterflies (leftmost example) and the fruit (example in the middle), COUNTGD correctly outputs a count of 0. For the flowers (rightmost example), COUNTGD incorrectly outputs a count of 9.

## H   Additional Qualitative results

In Figures 8 and 9 we include additional qualitative results from the CountBench [39] and FSC-147 [42] test sets respectively.

## I   Limitations

**Text is sometimes not enough to specify the object to count.**   Sometimes, the object to count looks uncommon and is so unique that text alone does not provide enough information to specify the object to count. For example, in Fig. 10 (b), providing the text "crystals" results in CountGD estimating an incorrect count of 2, while providing the text "crystals" together with one visual exemplar results in CountGD estimating a more accurate count in Fig. 10 (c). This happens because the crystals in the X-ray image in Fig. 10 (b) and (c) do not look like regular crystals such as those in Fig. 10 (a), so it is hard for CountGD to pick them out given only text. Providing the exemplar alleviates the issue, allowing CountGD to match the crystals in the X-ray image visually instead of relying on text alone.

**Very fine-grained counting can be challenging.**   CountGD sometimes struggles to count different categories of objects with text if the categories are very similar. For example, in Fig. 11, CountGD cannot pick out the baby penguins from the adult penguins. This is because the baby penguins and adult penguins look very similar in terms of color and shape.

## J   Broader Impacts

In general, object counting is an important task with many real-world applications. A strong counting model has positive impacts on various domains such as agriculture, satellite images, microscopy and medical images. However, there are also possible negative impacts, like violating privacy in human counting for surveillance cameras, or being used for military applications. This paper focuses on

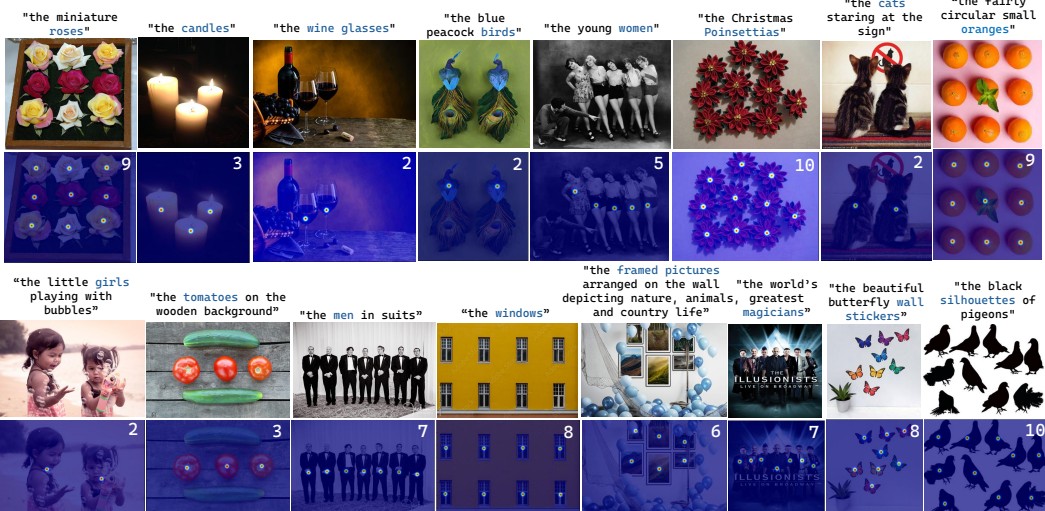

Figure 8: CountBench [39] counting examples using the multi-modal CountGD. The model is trained on visual exemplars and text from FSC-147 [42] and tested zero-shot with text only on CountBench. Blue letters indicate the subject of each caption input to the model. Detected points are filtered with a Gaussian and plotted under input images for visualization purposes. CountGD predicts the count in all the images shown with 100% accuracy. **Note** how in the top row CountGD correctly only counts the women, not the men, in "the young women" example, and only the alive cats (not the one painted on the wall) in "the cats starting at a wall". In the bottom row, the model also correctly does *not* count the repeated bubbles near the little girls or the multiple balloons.

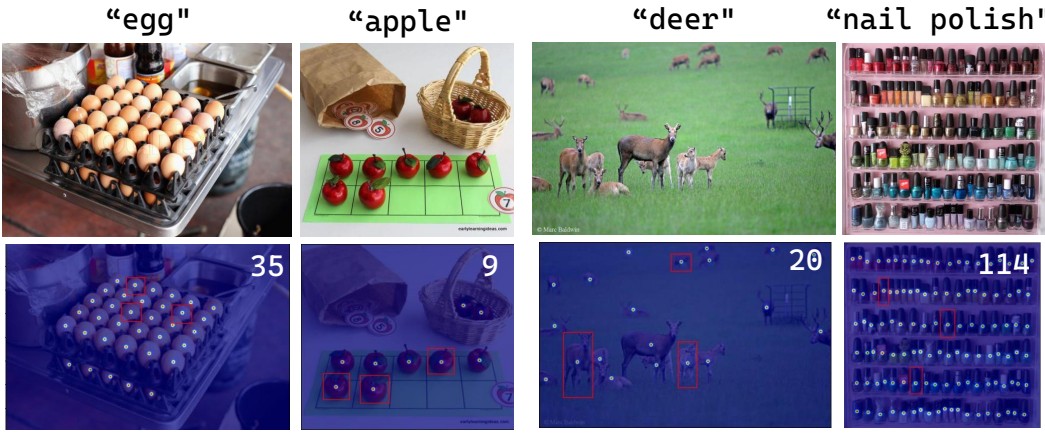

Figure 9: Additional qualitative examples showing CountGD's performance on the FSC-147 test set. In these examples, CountGD predicts the count with 100 % accuracy.

building a more accurate and flexible counting model. The broader impacts in real-world scenarios should be considered carefully before deploying the model.

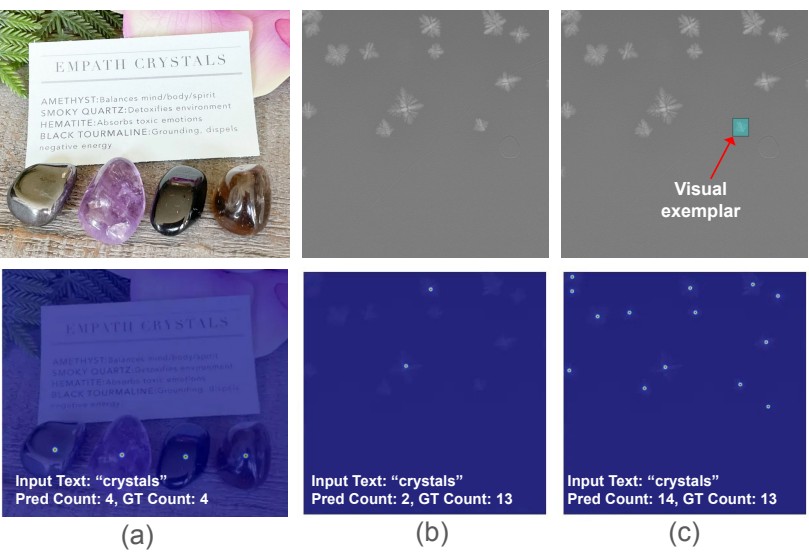

Figure 10: **Text is sometimes not enough to specify the object to count.** In (a), given only text, CountGD accurately estimates the number of crystals. In (b), CountGD cannot accurately estimate the number of crystals in the X-ray image using text alone, since they look unfamiliar. In (c), providing an additional visual exemplar alleviates the issue. Input images are in the top row. Detected instances from CountGD are shown in the bottom row.

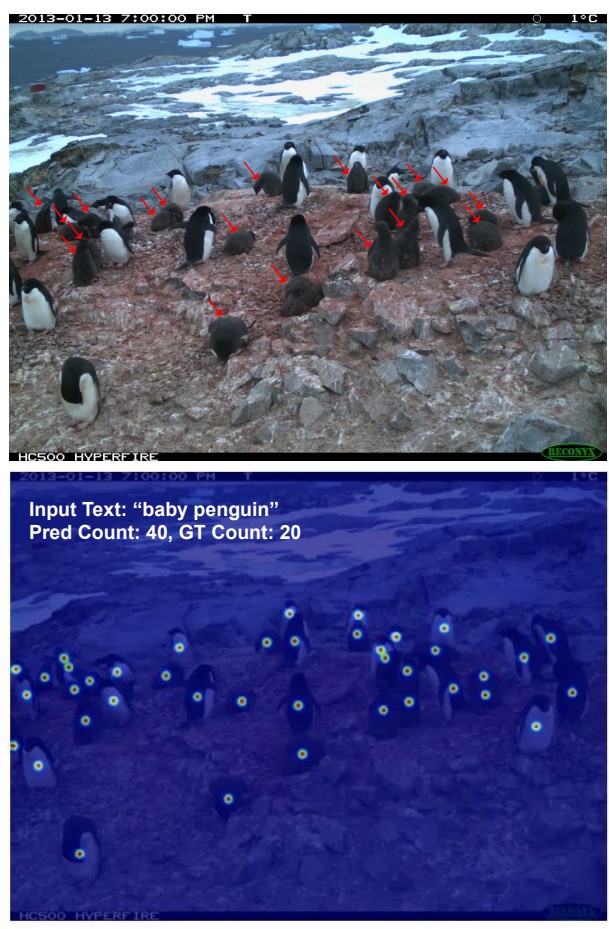

Figure 11: **Very fine-grained counting can be challenging.** CountGD cannot distinguish between the baby penguins (pointed to with red arrows in the top image) and the adult penguins. Given the text "baby penguin," CountGD counts all of the penguins in the input image. The adult and baby penguins look very similar. They have similar colors (mostly black) and shapes. The input image is in the top row. Detected instances from CountGD are shown in the bottom row.

