# OpenReview forum: "CountGD: Multi-Modal Open-World Counting"
_NeurIPS.cc/2024/Conference — NeurIPS 2024 poster_

### Official Review · Reviewer_44bk · 2024-07-12

**Soundness:** 3
**Presentation:** 3
**Contribution:** 3
**Rating:** 5
**Confidence:** 4

**Summary:**

This paper focuses on multi-modality open-world object counting, where the model can receive text or visual exemplars or both as input. To this end, the authors repurpose Grounding DINO, an open-vocabulary object detector, as an open-world object counting model. The main idea is to treat visual exemplars as additional tokens, which enables the interaction between exemplars and image/text features. Experiments on the FSC-147 dataset show that the proposed method achieves strong counting performance.

**Strengths:**

1. The idea of open-world object counting, where text and/or visual exemplars can be received as input, is promising in real-world applications.
2. This paper presents a simple way to repurpose Grounding DINO into an object counting model by casting visual exemplars as text tokens.
3. The proposed method delivers strong counting performance on the FSC-147 dataset.

**Weaknesses:**

1. Inaccurate claim. The authors claim that they introduce the first open-world counting model in the sense that the prompt can be specified by a text description or visual exemplars or both. However, PseCo [1] is also an open-world counting model that can receive text and/or visual exemplar as input.
2. It is unclear whether the proposed method can properly handle dense objects. As mentioned on Page 5, the authors set the number of object queries to 900, which implies that the maximum number of predicted objects is 900. This is acceptable in object detection but not satisfactory in object counting. In real-world applications, the number of objects may exceed 1000. It is suggested to divide images into different groups based on the number of objects, and report detailed results on these groups. This can help the readers understand how the proposed method performs under different object densities.
3. The comparisons in Table 1 are somewhat unfair. As mentioned in Appendix D, the authors use test-time normalization to alleviate double counting. Without this technique, the proposed method does not show superiority over existing methods. In addition, deploying adaptive cropping significantly improves the accuracy on the test set of FSC-147. This echoes the concern regarding the capability of CountGD to tackle dense objects.
4. It appears that the proposed method can not output bounding boxes. This undermines the value of the proposed open-world counting model. As Grounding DINO is an open-set object detector, one would expect the repurposed object counting model can localize objects. For comparison, DAVE [35] and PseCo [1] can output object boxes.
5. Missing comparisons with baseline Grounding DINO. The authors propose to finetune the feature enhancer and cross-modality decoder to achieve open-world object counting. Therefore, it is necessary to compare CountGD with original Ground DINO to support the effectiveness of such operations. If Grounding DINO already excels in object counting, there is no need to finetune the model.

Reference:

[1] Point, Segment, and Count: A Generalized Framework for Object Counting. CVPR 2024.


Minor issues:
* A few methods are not properly cited. For example, DAVE [35] is published in CVPR 2024 and LOCA [9] is published in ICCV 2023. The authors should properly cite previous works.
* The ``Published`` column in Table 1 provides little information about the compared methods. It is suggested to replace this column with paper venue.

**Questions:**

1. Can the proposed method output object boxes? Considering that Ground-DINO is an open-world object detector, it would be better if CountGD could also localize objects.
2. Did the authors evaluate Grounding DINO on the FSC-147 dataset?
3. Is the proposed method sensitive to confidence threshold sigma?

**Limitations:**

The paper mentions the potential limitation of counting error estimation.

---

> ### Author Rebuttal · Authors · 2024-08-05
>
> Thank you Reviewer 44bk for recognizing the promising real-world applications of our work and the strong performance of CountGD on object counting. We address all weaknesses (W1-W5), minor issues (M1-M2), and questions (Q1-Q3) below.
>
> **W1. Inaccurate claim. The authors claim that they introduce the first open-world counting model in the sense that the prompt can be specified by a text description or visual exemplars or both. However, PseCo [1] is also an open-world counting model that can receive text and/or visual exemplar as input.**\
> The reviewer's weakness is not valid for two reasons: (1) Our method, CountGD, is the first to be able to use both visual exemplars and text to specify the prompt, rather than using each independently; (2) PseCo was published at CVPR 2024, after the NeurIPS submission deadline. In more detail, CountGD is the first open-world counting method for which prompting with both visual exemplars and text is studied and leveraged explicitly. CountGD *learns* to fuse the exemplars and text using self-attention to achieve better performance than *both* exemplar-based alone and text-based alone approaches. In contrast, PseCo does not learn to relate the visual exemplars to the text, and does not achieve better performance than methods that ingest one modality at a time. For example, PseCo does not perform better than prior exemplar-based method LOCA [9]. While PseCo can accept either text inputs or exemplars, it is not evaluated on the case where both are provided simultaneously. PseCo performs far worse than CountGD as shown in Tab. 3 of the rebuttal. This demonstrates that PseCo is not designed to effectively leverage *both* exemplars and text to achieve better performance than when only exemplars or only text is provided. Finally, as mentioned before, PseCo was published in CVPR 2024 after the NeurIPS deadline.
>
> **W2. It is unclear whether the proposed method can properly handle dense objects. As mentioned on Page 5, the authors set the number of object queries to 900, which implies that the maximum number of predicted objects is 900.**\
> CountGD can accurately count given images with greater than 900 objects. It does this using the adaptive cropping scheme detailed in App D. This overcomes a limitation of previous methods that use a fixed number of queries, e.g. GroundingDINO, and is another original contribution of the paper. As requested, we split images into different groups according to the number of objects they contain, and report the percent error for each group. We find that for FSC-147 test images with more than 900 objects, the mean percent error ($|\frac{gt - pred}{gt}|$x100%) is 10%, and for images with at most 900 objects, it is 8%. This shows that CountGD works well, even for images with greater than 900 objects.
>
> **W3. The comparisons in Tab. 1 are somewhat unfair. As mentioned in Appendix D, the authors use test-time normalization to alleviate double counting. Without this technique, the proposed method does not show superiority over existing methods. In addition, deploying adaptive cropping significantly improves the accuracy on the test set of FSC-147. This echoes the concern regarding the capability of CountGD to tackle dense objects.**\
> Prior method CounTR [27] also uses test-time normalization for its published results, so our comparisons are not unfair. Also, we already give results with and without the test-time normalization and without applying adaptive cropping in Tab. 4 of the Appendix. By adding adaptive cropping back in, we show CountGD achieves state-of-the-art counting accuracy even without the test-time normalization in Tab. 4 of the rebuttal. Furthermore, on CARPK and CountBench, no test-time normalization was applied, and CountGD still significantly beats the state-of-the-art approaches (Tab. 2 of the main paper). We already address counting in dense scenes in our response to W2.
>
> **W4. It appears that the proposed method can not output bounding boxes. This undermines the value of the proposed open-world counting model. As Grounding DINO is an open-set object detector, one would expect the repurposed object counting model can localize objects. For comparison, DAVE [35] and PseCo [1] can output object boxes.**\
> We do not understand why not outputting bounding boxes "undermines the value of the proposed open-world counting model." Our model (1) counts more accurately than PseCo and DAVE, and (2) localizes each object by outputting its center location. We are not proposing an object detector.
>
> **W5. Missing comparisons with baseline Grounding DINO.**\
> Thank you for this very good baseline suggestion. We find that CountGD is significantly more accurate at object counting than GroundingDINO. We give results for this baseline in Tab. 2 of the rebuttal and will include it in the final form of the paper.
>
> **M1: A few methods are not properly cited.**\
> Thank you for the helpful feedback on the citations. We will correct them for the camera ready.
>
> **M2. The "Published" column in Tab. 1 provides little information about the compared methods. It is suggested to replace this column with paper venue.**\
> We will replace the “Published” column in Tab. 1 with the paper venue. The point of the column was to be explicit that we were also including unpublished arXiv papers in our comparisons in the submission.
>
> **Q1. Can the proposed method output object boxes? Considering that GroundDINO is an open-world object detector, it would be better if CountGD could also localize objects.**\
> We already address this in our response to W4.
>
> **Q2. Did the authors evaluate Grounding DINO on the FSC-147 dataset?**\
> Yes, we have added these results in our response to W5 showing that CountGD is significantly better at object counting than GroundingDINO.
>
> **Q3. Is the proposed method sensitive to confidence threshold sigma?**\
> Yes, Appendix C says we pick sigma=0.23 out of {0.14,0.17,0.2,0.23,0.26} achieving min. val. MAE 7.1. The max. MAE of 9.5 was at sigma=0.14.

---

> > ### Comment · Reviewer_44bk · 2024-08-10
> > **Further questions**
> >
> > Thanks for the response. The rebuttal has addressed most of my concerns. I have two more questions.
> > 1. Is the model sensitive to input text descriptions? The authors mention that they use text descriptions from FSC-147-D [1]. What about using the class names provided by FSC-147 as input text descriptions?
> > 2. Is it possible to train the model with text and visual exemplars, but inference with only one modality? I understand that the authors focus on multi-modality open-world counting, but text and visual exemplars may not be available simultaneously in real-world applications. While it is feasible to deploy two different counting models (i.e., text-only and exemplar-only models), it would be better if the model could handle different inputs.

---

> > > ### Author Response · Authors · 2024-08-11
> > > **Addressing additional questions about FSC-147 class names and multi-modal inference**
> > >
> > > Thank you for the questions and for recognizing that we have addressed most of your concerns. We respond to each question below:
> > >
> > > **RE: Is the model sensitive to input text descriptions? The authors mention that they use text descriptions from FSC-147-D [1]. What about using the class names provided by FSC-147 as input text descriptions?**
> > >
> > > No, CountGD is **not** very sensitive to the text descriptions. By using exemplars and the class names in FSC-147 instead of the text in FSC-147-D, we achieve a 6.09 MAE and a 30.84 RMSE on FSC-147 Test, which is still state of the art. The original result using exemplars and the text in FSC-147-D is a 5.74 MAE, and a 24.09 RMSE.
> > >
> > > **RE: Is it possible to train the model with text and visual exemplars, but inference with only one modality? I understand that the authors focus on multi-modality open-world counting, but text and visual exemplars may not be available simultaneously in real-world applications. While it is feasible to deploy two different counting models (i.e., text-only and exemplar-only models), it would be better if the model could handle different inputs.**
> > >
> > > **Yes**, inference with one modality is possible with our multi-modal model trained with text and visual exemplars. We already show results for training on *both* text and visual exemplars and testing with *only* text on the CARPK dataset (Tab. 2, row 4 of the main paper) and the CountBench dataset (Tab. 2, row 10 of the main paper). In both cases, CountGD, trained with both text and visual exemplars, achieves state-of-the-art accuracy for open-world counting when given only text at inference. The same model given *only* visual exemplars at inference achieves 5.22 MAE and 7.14 RMSE on CARPK Test. This performance is slightly better than the visual exemplar only method CounTR [27] which was fine-tuned on CARPK. Note, our model is not fine-tuned on CARPK or CountBench.

---

> > > > ### Comment · Reviewer_44bk · 2024-08-12
> > > >
> > > > I think the authors have well addressed the concerns. I would like to raise my rating.

---

### Official Review · Reviewer_QCsU · 2024-07-12

**Soundness:** 3
**Presentation:** 4
**Contribution:** 2
**Rating:** 4
**Confidence:** 4

**Summary:**

The authors propose a novel object counting method CountGD that is based on GroundingDINO. It uses the strong localization and multimodal capabilities of GroundingDINO and adapts it to the task of few-shot object counting. The main architectural change is the introduction of visual exemplars to the GroundingDINO architecture by ROI pooling exemplar boxes and using the resulting features as additional visual tokens. The proposed method, CountGD, achieves excellent performance on the standard FSC-147 few-shot counting task and excels in the text-condition counting scenario, significantly outperforming the previous state-of-the-art. Additionally, it can utilize both visual exemplars and text conditioning simultaneously.

**Strengths:**

- The paper is clearly written and the method is presented well.
- The main part of the evaluation is thorough, comparing the proposed CountGD on the standard FSC-147 dataset on both the few-shot setup using visual exemplars and the text-conditioned setup.
- The appendix contains a good amount of technical details that help with method understanding and implementation.
- CountGD achieves state-of-the-art results on the FSC-147 and CountBench datasets.

**Weaknesses:**

- The qualitative comparison is a bit confusing. The output of the method is visualized as a density map for example in Figure 4. However, CountGD outputs a similarity score for each selected query that is then thresholded to classify which queries correspond to either exemplar objects or text queries. The visualization process must therefore be hand crafted and is not a density map, so the visualization is a bit misleading.

- The method is based heavily on GroundingDINO and is trained on object counting datasets following the training process of previous few-shot counting works. The performance is additionally boosted due to the post processing approaches, several of which have been proposed in previous works. While the combination of GroundingDINO and contributions from previous counting works are interesting, and the performance of CountGD is strong, the conceptual contributions presented in the paper are few and unconvincing.

- There are some clarity issues in terms of the training setup (Question 1).
- The ablation study in the main paper could be expanded as currently only the impact of training/inferring with different modalities is evaluated.

**Questions:**

- In Eq. 3, the L_loc is the L1 loss between ground truth centers and the predicted bounding box centers. So CountGD also outputs bounding boxes? This is not evaluated anywhere although methods like DAVE use benchmarks such as FSCD-147 for object detection performance evaluation.
- Have you run ablation experiments for individual components in the loss function?
- Have you performed experiments evaluating the generalization capability with a single exemplar? A one-shot experiment might be interesting given the strong performance in the few-shot scenario.

- The contribution over the standard GroundingDINO architecture seems minor. Essentially, the output queries of GroundingDINO are compared to the extracted exemplar features (text+visual exemplars) and the similarity is then thresholded to obtain the detections. While this is a simple solution that seems to work well it does not seem to give a significant insight into solving conditional object counting and gives the impression that it is merely relying on the very strong features produced by GroundingDINO. A further discussion on why the proposed concepts presented in the paper are a major contribution would be beneficial. This is my main issue and the main reason for my score.

**Limitations:**

The authors have addressed the limitations.

---

> ### Author Rebuttal · Authors · 2024-08-05
>
> Thank you Reviewer QCsU for recognizing our experiments are thorough, our results significantly improve the state-of-the-art, and our work is presented well. We address all weaknesses (W1-W4) and questions (Q1-Q4) below. We combine some weaknesses and questions if they make overlapping points.
>
> **W2. The method is based heavily on GroundingDINO and is trained on object counting datasets following the training process of previous few-shot counting works.**\
> **Q4. The contribution over the standard GroundingDINO architecture seems minor. Essentially, the output queries of GroundingDINO are compared to the extracted exemplar features (text+visual exemplars) and the similarity is then thresholded to obtain the detections. While this is a simple solution that seems to work well it does not seem to give a significant insight into solving conditional object counting and gives the impression that it is merely relying on the very strong features produced by GroundingDINO. A further discussion on why the proposed concepts presented in the paper are a major contribution would be beneficial. This is my main issue and the main reason for my score.**\
> We combine W2 and Q4, since they raise similar concerns. We respond on these points first as the reviewer says "This is my main issue and the main reason for my score."
>
> We do benefit from the strong features from GroundingDINO (that is our design choice), but we go beyond just comparing the queries of GroundingDINO to the text + visual exemplar features. CountGD uses self-attention between the visual exemplars and the text inside the feature enhancer to learn to fuse them together earlier, before the queries are constructed.  In fact, the queries produced by CountGD are different from the queries output by GroundingDINO due to cross-attention between the fused visual exemplar and text features and image features in the feature enhancer. CountGD also achieves far better accuracy than all prior approaches, including GroundingDINO, as shown in Tab. 2 of the rebuttal and Tab. 1 and 2 of the main paper. This is a significant contribution because GroundingDINO and prior counting methods were not able to effectively leverage both the exemplars and the text simultaneously to achieve better counting accuracy than methods that ingest one modality at a time. Our approach to *learn* to relate the visual exemplars to the text with self-attention accomplishes this for the first time and has not been proposed by prior work.
>
> Furthermore, the proposed SAM-based approach for handling self-similarity detailed in Appendix D is completely novel and works for cluttered scenes, which is not true for the TT-Norm proposed by the prior counting method CounTR. Finally, no prior work investigates interactions between visual exemplars and text, which is helpful for filtering instances detected with exemplars based on attributes such as color and location. In summary, CountGD is the first work to comprehensively address the multi-modal counting setting where both exemplars and text are available.
>
> **W1. The qualitative comparison is a bit confusing. The output of the method is visualized as a density map for example in Fig. 4. However, CountGD outputs a similarity score for each selected query that is then thresholded to classify which queries correspond to either exemplar objects or text queries. The visualization process must therefore be hand crafted and is not a density map, so the visualization is a bit misleading.**\
> None of the visualizations for the qualitative examples are handcrafted. As specified in the figure captions, all plots were generated automatically from the center points predicted by CountGD. In Fig. 4, 7, and 8 in the main paper, we place a 1 on the center points and filter them with a Gaussian so they are easier to see for readers. In Fig. 5, we plot the similarity (confidence) scores at the predicted center locations to illustrate the interaction between the exemplars and the text (e.g. specifying “red” increases the confidence of the red object).
>
> **W3. There are some clarity issues in terms of the training setup (Q1).\
> Q1. In Eq. 3, the L_loc is the L1 loss between ground truth centers and the predicted bounding box centers. So CountGD also outputs bounding boxes? This is not evaluated anywhere although methods like DAVE use benchmarks such as FSCD-147 for object detection performance evaluation.**\
> We combine W3 and Q1, since W3 refers directly to Q1. CountGD outputs object center locations, not bounding boxes. This is so that the output of CountGD aligns with the training data in FSC-147, which only provides object centers, when training.  FSC-D147 only provides bounding boxes for the validation and test sets, not the training set. In Eq. 3, L_loc is between the center points predicted by CountGD and the ground truth object centers provided by FSC-147. DAVE, a model that performs well and outputs bounding boxes, is concurrent work. DAVE became available on arXiv less than one month before the NeurIPS submission deadline and was published in CVPR afterwards.
>
> **W4. The ablation study in the main paper could be expanded as currently only the impact of training/inferring with different modalities is evaluated.**\
> In addition to an ablation on training/inferring with different modalities in the main paper, we also provide an ablation on post-processing techniques in Tab. 4 of the Appendix.
>
> **Q2. Have you run ablation experiments for individual components in the loss function?**\
> Yes, in Appendix C, we test (lambda_loc,lambda_cls) in {1,2.5,5}x{1,2.5,5} and pick values that achieve the best validation MAE. We include a sensitivity test in Tab. 5 of the rebuttal for this.
>
> **Q3. Have you performed experiments evaluating the generalization capability with a single exemplar? A one-shot experiment might be interesting given the strong performance in the few-shot scenario.**\
> We provide the results for this experiment in Tab. 1 of the rebuttal.

---

> > ### Comment · Area_Chair_AT8L · 2024-08-11
> > **Has the rebuttal addressed your concerns?**
> >
> > Dear Reviewer QCsU,
> >
> > Thank you again for your time to review this paper. Could you please check if the authors' rebuttal has addressed your concerns at your earliest convenience? Thank you!
> >
> > Best regards,
> >
> > AC

---

> > ### Comment · Reviewer_QCsU · 2024-08-13
> >
> > Thanks for the detailed rebuttal, it has addressed most of my concerns. I still, however am not entirely convinced about the performance of Grounding DINO on the FSC147 dataset. The authors provide the results of a pretrained Grounding DINO directly applied to FSC147, however this has not been optimized to handle the distribution of objects in the FSC147 dataset at all. In the Reffering Expression Counting paper[1] an experiment was performed, where Grounding DINO was finetuned on the FSC147 dataset. Table 4 in the paper[1]. While conditioned on text only, a finetuned Grounding DINO achieved results that match the performance of CountGD. A comment on these results would be greatly appreciated. Also I am aware that Reffering Expression Counting is concurrent work and I am not asking for a comparison to the proposed method, just a comment on the surprisingly good performance of a finetuned Grounding DINO.
> >
> > [1] https://openaccess.thecvf.com/content/CVPR2024/papers/Dai_Referring_Expression_Counting_CVPR_2024_paper.pdf

---

> > > ### Author Response · Authors · 2024-08-13
> > > **Adding a Comment on Fine-Tuned GroundingDINO**
> > >
> > > Thank you for the suggestion and for recognizing that we have addressed most of your concerns. We will add the following comment to the “FSC-147” paragraph of Section 4.3 of the camera-ready as well as a new citation to the Referring Expression Counting [1] paper.
> > >
> > > *While a pre-trained GroundingDINO performs poorly at counting, a GroundingDINO model fine-tuned on FSC-147 achieves good results [1], that match the performance of CountGD when trained and tested with text only. Adding visual exemplars to CountGD significantly improves its performance over fine-tuned GroundingDINO (main paper, Table 1, lowest row shows a test MAE of 5.74 and a test RMSE of 24.09 for multi-modal CountGD compared to the test MAE of 10.82 and test RMSE of 104 noted in [1] for fine-tuned GroundingDINO). Unlike CountGD, GroundingDINO does not allow for visual exemplars as additional inputs.*
> > >
> > > [1] [https://openaccess.thecvf.com/content/CVPR2024/papers/Dai_Referring_Expression_Counting_CVPR_2024_paper.pdf](https://openaccess.thecvf.com/content/CVPR2024/papers/Dai_Referring_Expression_Counting_CVPR_2024_paper.pdf)

---

### Official Review · Reviewer_Jn2T · 2024-07-12

**Soundness:** 3
**Presentation:** 3
**Contribution:** 4
**Rating:** 7
**Confidence:** 4

**Summary:**

This paper aims to improve open-vocabulary object counting in images by repurposing an existing detection model (GroundingDINO) and introducing multi-modal prompts using text descriptions and visual exemplars. The contributions include the introduction of COUNTGD, the first open-world counting model, improved performance on counting benchmarks, and a preliminary study on the interaction between text and visual prompts.

**Strengths:**

1. The paper introduces a new task setting called multi-modal open-world counting, where counting prompts can be specified through text, visual exemplars, or a combination of both. This multi-modal approach significantly enhances practicality and interactivity in object counting tasks.
2. The proposed single-stage model, COUNTGD, unifies the previous approach of using either visual exemplars or text as prompts into a single framework, achieving state-of-the-art counting performance.
3. Extensive dataset validations and comparisons with multiple methods demonstrate the generality and effectiveness of the proposed approach.

**Weaknesses:**

It would be beneficial to delve deeper into the interaction between text and visual exemplars in research. For example, studying the impact of factors such as the ratio of trainable text to visual exemplars on the model's performance.

**Questions:**

1. Could the author provide some case studies and analysis of current model failures in the text or as an appendix?
2. The NeurIPS requirement states that the abstract must be limited to one paragraph.
3. What would be the result when there is a conflict between visual exemplar and text?

**Limitations:**

Yes

---

> ### Author Rebuttal · Authors · 2024-08-05
>
> Thank you Reviewer Jn2T for recognizing the extensiveness of our experiments, the novelty of our multi-modal counting setting, and how our approach significantly improves the practicality and accuracy of object counting. We address all weaknesses (W1) and questions (Q1-Q3) below.
>
> **W1. It would be beneficial to delve deeper into the interaction between text and visual exemplars in research. For example, studying the impact of factors such as the ratio of trainable text to visual exemplars on the model’s performance.**\
> This is an interesting question, and has several aspects. First, we consider how changing the ratio of text to visual exemplars in the prompt *during inference* influences model performance. As shown in Table 1 of the rebuttal, we find that after providing a short text input, increasing the number of exemplars in the prompt increases the accuracy of the model. On the other hand, increasing the *length* of the text only improves the accuracy if the *content* of the text is more informative about the object, such as specifying its color and shape.
>
> Second, we will train with different ratios of visual exemplars to text before the camera ready deadline, and investigate how this influences model performance. But there is insufficient time to retrain during the rebuttal period.
>
> **Q1. Could the author provide some case studies and analysis of current model failures in the text or as an appendix?**\
> Yes, in addition to the limitations we discuss in Appendix F, we will add the 2 case studies and analysis of model failures that we provide below to the camera ready.
>
> 1. *Text is sometimes not enough to specify the object to count.* \
> Sometimes, the object to count looks uncommon and is so unique that text alone does not provide enough information to specify the object to count. For example, in Fig. 1 (b) of the rebuttal, providing the text “crystals" results in CountGD estimating an incorrect count of 2, while providing the text “crystals" together with one visual exemplar results in CountGD estimating a more accurate count in Fig. 1 (c). This happens because the crystals in the X-ray image in Fig.1 (b) and (c) do not look like regular crystals such as those in Fig. 1 (a), so it is hard for CountGD to pick them out given only text. Providing the exemplar alleviates the issue, allowing CountGD to match the crystals in the X-ray image visually instead of relying on text alone.
> 2. *Very fine-grained counting can be challenging.*\
> CountGD sometimes struggles to count different categories of objects with text if the categories are very similar. For example, in Fig. 2 of the rebuttal, CountGD cannot pick out the baby penguins from the adult penguins. This is because the baby penguins and adult penguins look very similar in terms of color and shape.
>
> **Q2. The NeurIPS requirement states that the abstract must be limited to one paragraph.**\
> We will remove the line breaks so that there is a single paragraph for the camera-ready version.
>
> **Q3. What would be the result when there is a conflict between visual exemplar and text?**\
> Following the procedure in Section 4.5, when there is a conflict between the visual exemplar and the text, the model counts neither the objects that match the exemplar, nor the objects that match the text. This is because objects need to match the aggregation of both the exemplar and the text features to be counted, which might not carry meaningful information when the exemplar and text contradict. For example, if a user specifies an exemplar of a blueberry with the text “strawberry'' for an image with both blueberries and strawberries, CountGD will output 0, since no object in the image is *both* a blueberry and a strawberry. Furthermore, the inference procedure in Section 4.5 can be modified such that objects that match the exemplar *or* the text are counted, in which case *both* the blueberries and the strawberries would be counted.

---

> > ### Comment · Reviewer_Jn2T · 2024-08-08
> >
> > Regarding the response to Q3, I hold a cautious attitude of skepticism. The model may not necessarily capture conflicts between text and visual examples effectively, and the article does not explicitly constrain this aspect. Therefore, confusion is highly likely to occur. More real-world testing examples need to be provided and analyzed. Of course, I understand that the authors may not be able to provide more images at this point, so presenting results and analysis of test examples would be acceptable. More importantly, I hope that the author can explore this issue carefully. If necessary, it is best to provide relevant result cases and analysis in the camera-ready version, which will help others to evaluate the article and further study.

---

> > > ### Author Response · Authors · 2024-08-09
> > > **Addressing conflicts between text and visual exemplars**
> > >
> > > We appreciate the skepticism. As we explained in the rebuttal, our ideal output for a conflict would be zero - i.e. we consider the AND operation between the text and exemplars. We investigate to what extent this holds in practice by defining three levels of conflict.
> > >
> > > super-class conflict -- the super class of the exemplar and the text don't match e.g., the exemplar is a tiger and the text=’chair’;
> > >
> > > sub-class conflict -- the sub-class of the exemplar and text don't match e.g., the exemplar is a man and the text=’woman’, both of which are humans;
> > >
> > > attribute conflict -- the exemplar and text match in terms of class but don't match in terms of attribute, e.g., the exemplar is a blue circle but the text='red'
> > >
> > > For case (1) we use the image of the butterflies in Fig. 7 of the main paper. By providing visual exemplars of the butterflies and the text ‘white plant pot,’ we get a count of 0 as expected. For case (2) we use the image of the strawberries and blueberries in Fig. 1 (b) of the main paper. By providing one exemplar of a blueberry and the text ‘strawberry, we obtain a count of 0. For case (3), we consider the colored roses in Fig. 4. In this case, when providing an exemplar of a red rose and the text ‘yellow,’ the output is (incorrectly) 9, the number of red and yellow roses. We speculate that we are limited by image-text capabilities of the original GroundingDINO model (as illustrated in the fine-grained limitation example provided to Reviewer Jn2T).  We will include the output of CountGD in these cases and detailed analysis in the camera-ready.

---

> > > > ### Comment · Reviewer_Jn2T · 2024-08-11
> > > >
> > > > Thanks for responses. Keep my ratings. Good luck!

---

### Author Rebuttal · Authors · 2024-08-05

We provide the figures and tables for the rebuttal as a PDF attached here.

---

### Comment · Area_Chair_AT8L · 2024-08-09

Dear Reviewers,

Thank you very much again for your valuable service to the NeurIPS community.

As the authors have provided detailed responses, it would be great if you could check them and see if your concerns have been addressed if you haven't done so. Your prompt feedback would provide an opportunity for the authors to offer additional clarifications if needed.

Best regards,

AC

---

### Decision · Program_Chairs · 2024-09-25

**Decision:**

Accept (poster)

**Comment:**

Both reviewers Jn2T and 44bk provided positive reviews for this paper, recognizing the significance of the new open-world counting, where counting prompts can be specified through text, visual exemplars, or a combination of both. The proposed approach, based on GroundingDINO, achieves strong empirical results, where extensive dataset validations and comparisons with multiple methods are reported.

Initially, the reviewer QCsU shared concerns about the paper, which have been addressed after the user rebuttal period. Though the reviewer still put a negative review score (Borderline Reject), according to the discussions between the authors and reviewer, no substantial concerns remain.

The AC therefore recommends to accept this paper. The authors are highly encourage to incorporate the new results reported in the rebuttal into the final version of the paper.